# A Double-Deck Structure of Reduced Graphene Oxide Modified Porous Ti_3_C_2_T_x_ Electrode towards Ultrasensitive and Simultaneous Detection of Dopamine and Uric Acid

**DOI:** 10.3390/bios11110462

**Published:** 2021-11-18

**Authors:** Yangguang Zhu, Qichen Tian, Xiufen Li, Lidong Wu, Aimin Yu, Guosong Lai, Li Fu, Qiuping Wei, Dan Dai, Nan Jiang, He Li, Chen Ye, Cheng-Te Lin

**Affiliations:** 1Laboratory of Environmental Biotechnology, School of Environmental and Civil Engineering, Jiangnan University, Wuxi 214122, China; zhuyangguang@nimte.ac.cn; 2Key Laboratory of Marine Materials and Related Technologies, Zhejiang Key Laboratory of Marine Materials and Protective Technologies, Ningbo Institute of Materials Technology and Engineering (NIMTE), Chinese Academy of Sciences, Ningbo 315201, China; daidan@nimte.ac.cn (D.D.); jiangnan@nimte.ac.cn (N.J.); lihe@nimte.ac.cn (H.L.); 3College of Chemical Engineering, Zhejiang University of Technology, Hangzhou 310032, China; tianqichen@nimte.ac.cn; 4Key Laboratory of Control of Quality and Safety for Aquatic Products, Chinese Academy of Fishery Sciences, Beijing 100141, China; wulidong19849510@hotmail.com; 5Department of Chemistry and Biotechnology, Faculty of Science, Engineering and Technology, Swinburne University of Technology, Hawthorn, VIC 3122, Australia; aiminyu@swin.edu.au; 6Department of Chemistry, Hubei Normal University, Huangshi 435002, China; gslai@hbnu.edu.cn; 7College of Materials and Environmental Engineering, Hangzhou Dianzi University, Hangzhou 310018, China; fuli@hdu.edu.cn; 8School of Materials Science and Engineering, Central South University, Changsha 410083, China; qiupwei@csu.edu.cn; 9Center of Materials Science and Optoelectronics Engineering, University of Chinese Academy of Sciences, Beijing 100049, China

**Keywords:** reduced graphene oxide, Ti_3_C_2_T_x_, dopamine, uric acid, double deck

## Abstract

Considering the vital physiological functions of dopamine (DA) and uric acid (UA) and their coexistence in the biological matrix, the development of biosensing techniques for their simultaneous and sensitive detection is highly desirable for diagnostic and analytical applications. Therefore, Ti_3_C_2_T_x_/rGO heterostructure with a double-deck layer was fabricated through electrochemical reduction. The rGO was modified on a porous Ti_3_C_2_T_x_ electrode as the biosensor for the detection of DA and UA simultaneously. Debye length was regulated by the alteration of rGO mass on the surface of the Ti_3_C_2_T_x_ electrode. Debye length decreased with respect to the rGO electrode modified with further rGO mass, indicating that fewer DA molecules were capable of surpassing the equilibrium double layer and reaching the surface of rGO to achieve the voltammetric response of DA. Thus, the proposed Ti_3_C_2_T_x_/rGO sensor presented an excellent performance in detecting DA and UA with a wide linear range of 0.1–100 μM and 1–1000 μM and a low detection limit of 9.5 nM and 0.3 μM, respectively. Additionally, the proposed Ti_3_C_2_T_x_/rGO electrode displayed good repeatability, selectivity, and proved to be available for real sample analysis.

## 1. Introduction

Dopamine (DA) is a catecholamine neurotransmitter in the central nervous system which contributes to various physiological functions, including memory, stimulus-response, motion control and vasodilation. [1,2]. The abnormality of DA is clinically related to several neurological disorders, such as senile dementia, Parkinson’s disease, and schizophrenia [3]. Uric acid (UA) is the major end product of purine metabolism, and an excess of UA levels may lead to serious chronic and metabolic diseases, such as gouty, hyperuricemia and kidney injury [4,5]. Considering the vital physiological functions of DA and UA and their coexistence in the biological matrix, the development of biosensing techniques for their simultaneous detection with high sensitivity is desirable for diagnostic and analytical applications [6,7].

Conventional analytical methods for the simultaneous detection of DA and UA, such as high-performance liquid chromatography (HPLC), chemiluminescene, and capillary electrophoresis, have been under development for decades [8,9,10]. As DA and UA are electrochemically active compounds, the electrochemical method has been adopted for the detection of these biomolecules with high sensitivity, simplicity and time efficiency [11,12,13]. However, the oxidation peak positions of these biomolecules are almost the same and difficult to distinguish effectively when using conventional electrodes such as glassy carbon electrodes (GCE) [14]. By using various nanomaterials modified on GCE chemically, the peak resolutions of these biomolecules have been much improved [15]. Therefore, this method has been widely adopted for the recognition of DA and UA simultaneously [16,17].

Among them, graphene has received extensive attention, due to its high surface-to-volume ratio, good electrical conductivity and high carrier mobility [18,19,20,21,22]. Kim et al. proposed a graphene-modified electrode for the selective detection of DA with a linear range of 4.0–100.0 μM and a detection limit of 2.6 μM [23]. Qi et al. constructed an electrochemical sensor based on pristine graphene to detect DA and UA, achieving a linear range of 5.0–710 μM and 6.0–1330 μM and a detection limit of 2.0 and 4.8 μM, respectively [24]. Gao et al. fabricated a graphene oxide (GO)-modified GCE with the covalent coupling method, indicating a good performance in sensing DA with a detection range of 1.0–15.0 μM and a detection limit of 0.27 μM [25]. However, since DA in human blood is usually low as 0.01–1 μM, the sensitivity of graphene-modified electrodes needs to be further improved [26]. Conventionally, using a graphene hybrid with metal (Au, Pt, Ag) nanoparticles (NPs) or carbon nanomaterial (as carbon nanotubes (CNTs)) is a common approach to increase the electrochemical activity of a modified electrode. Wang et al. synthesized novel Au NPs and reduced the graphene oxide (rGO) composite film by electrodepositing AuNPs onto the rGO surface, showing good performance in its ability to detect DA and UA with a linear range of 6.8–41.0 μM, 8.8–53.0 μM and a low detection limit of 1.4, 1.8 μM, respectively [27]. Sun et al. demonstrated a novel sensor based on graphene and Pt NPs nanocomposite by self-assembling Pt NPs onto the graphene surface, indicating its excellent performance in detecting DA and UA with a linear range of 0.03–8.13 μM, 0.05–11.9 μM and a low detection limit of 0.03, 0.05 μM, respectively [28]. Sun et al. developed a sensor based on CNTs and GO nanocomposite, exhibiting its performance in detecting DA and UA with a linear range of 5.0–500 μM, 3.0–60.0 μM and a low detection limit of 1.5, 1.0 μM, respectively [29].

Compared to the electrode modified by graphene, the detection performance of the electrode modified by graphene-based nanocomposite is improved, but it still needs to be further promoted. The interfacial binding strength of graphene-based nanocomposite and electrode may not be high either. To overcome this problem, a new and promising 2D nanomaterial with a 2D-layered structure, MXene, especially titanium carbide MXene (Ti_3_C_2_T_x_), has been extensively applied as a material with a high number of electric electrodes for batteries, supercapacitors and electrochemical detection [30,31,32,33]. Due to its excellent metallicity, electrical conductivities, hydrophilic surfaces, and environmental-friendly characteristics, Ti_3_C_2_T_x_ has been employed for the electrochemical detection of biomolecules, H_2_O_2_, and heavy metal ions [34,35,36]. Murugan et al. proposed a Ti_3_C_2_T_x_-modified electrode, which exhibited good performance in determining DA and UA and obtained a low detection limit of 0.06 and 0.08 μM, respectively [37]. These successful applications of Ti_3_C_2_T_x_ in the electrochemical detection prove that Ti_3_C_2_T_x_ is an ideal conductive matrix and improves electron transfer kinetic effectively. Particularly, the Ti-O-C covalent bonding is formed at the Ti_3_C_2_T_x_/rGO heterointerface via nucleophilic substitution dehydration reaction, and charge transport through the heterointerface is increased [38]. Therefore, the interfacial binding strength of Ti_3_C_2_T_x_/rGO heterointerface increases, resulting in an excellent electrochemical performance in detecting biomolecules. The Debye screening length, λ_D_, is defined as the effective thickness of the equilibrium double layer (EDL) [39]. The detection limit of biosensors is determined by λ_D_ between the surface of sensitive nanomaterials and the electrolyte [40]. Thus, λ_D_ can be altered effectively to obtain the low detection limit of biosensors based on the Ti_3_C_2_T_x_/rGO heterostructure.

In this work, we attempted to construct Ti_3_C_2_T_x_/rGO heterostructure with double-deck layer through electrochemical reduction. The rGO was modified on porous Ti_3_C_2_T_x_ electrode as the biosensor for the detection of DA and UA simultaneously. The Debye length was regulated by the alteration of rGO on the surface of Ti_3_C_2_T_x_ electrode. As evidenced by the differential pulse voltammetry (DPV) test, this proposed Ti_3_C_2_T_x_/rGO sensor exhibited an excellent performance in detecting DA and UA with a linear range of 0.1–100 μM and 1–1000 μM and a low detection limit of 0.0095 and 0.3 μM, respectively. Additionally, the proposed biosensor indicated good repeatability, selectivity, and potential for real sample analysis.

## 2. Materials and Methods

### 2.1. Chemicals

Potassium chloride (KCl), sodium chloride (NaCl), sodium sulphate (Na_2_SO_4_), dibasic sodium phosphate (Na_2_HPO_4_), potassium dihydrogen phosphate (KH_2_PO_4_), and Uric acid (UA) were purchased from Sinopharm Chemical Reagent Co., Ltd. (Shanghai, China). Dopamine (DA) was purchased from Shanghai Aladdin Biochemical Technology Co., Ltd. (Shanghai, China). All of the above chemical reagents were analytical reagents and were used without further purification. GO powder and rGO water dispersion were purchased from Nanjing JCNANO Technology Co., Ltd. (Nanjing, China). Ti_3_C_2_T_x_ water dispersion was purchased from Beike 2D materials Co., Ltd. (Suzhou, China). Deionized Milli-Q water (18.2 MΩ/cm) was used throughout the experiments.

### 2.2. Fabrication of Ti_3_C_2_T_x_/rGO Electrodes

The Ti_3_C_2_T_x_ water dispersion and GO water dispersion were dispersed ultrasonically for 1 h in an ice bath. Before modification, GCE electrodes with a diameter of 3 mm were polished using a 0.05 μm alumina slurry and cleaned in deionized water and ethanol by ultrasonication. Following that, GCE was activated via repetitive potential range scanning from −1–1 V with a scan rate of 0.1 V/s in 0.5 M H_2_SO_4_. The Ti_3_C_2_T_x_ dispersion was uniformly dropped onto the surface of the GCE and dried, followed by GO dispersion in the same way (Ti_3_C_2_T_x_/GO electrode). The Ti_3_C_2_T_x_/rGO-modified GCE was obtained through the electrochemical reduction method of immersing Ti_3_C_2_T_x_/GO into PBS with cyclic voltammetry (CV) sweeping in the potential range of 0.0–1.4 V at a scan rate of 0.1 V/s for 5 cycles, which was defined as the experimental group (Ti_3_C_2_T_x_/rGO electrode). As shown in Appendix A (see Appendix A), a large reduction peak was observed at the potential peak position of −1.23 V in the first cycle, and vanished subsequently, which referred to the electrochemical reduction process of GO to rGO. As controls, Ti_3_C_2_T_x_-modified GCE (Ti_3_C_2_T_x_ electrode) and rGO modified-GCEs (rGO electrode) were also prepared using the same method.

### 2.3. Characterizations

Field emission scanning electron microscope (FE-SEM QUANTA 250 FEG, FEI, Hillsboro, OR, USA) and energy dispersive spectroscopy (EDS)-mapping were applied to observe the morphology of the nanomaterials, including CNTs and AgNWs, separately, and the composite material. The surface compositions and chemical states were carried out by Raman spectroscopy (Renishaw in Via Reflex, Renishaw plc, Wotton-under-Edge, London, UK) with a laser wavelength of 532 nm and X-ray photoelectron spectroscopy (XPS, Axis Ultra DLD, Kratos Analytical, Manchester, UK), respectively. All electrochemical experiments were conducted with a CHI660e electrochemical workstation (Shanghai Chenhua Co., Ltd., Shanghai, China).

### 2.4. Electrochemical Tests

The electrolyte was a phosphate buffer solution (PBS) which contained 137 mM NaCl, 102.7 mM KCl, 8.1 mM Na_2_HPO_4_, and 1.8 mM KH_2_PO_4_ (pH ≈ 7.4). The bare GCE, Ti_3_C_2_T_x_, rGO, and Ti_3_C_2_T_x_/rGO were applied as working electrodes, which were separately immersed into PBS containing different DA concentrations, from 9.5 nM to 100 μM, and different UA concentrations, from 0.3 μM to 1000 μM, respectively, to compare their electrochemical performances. A saturated calomel electrode (SCE, Pt Hg(l)|Hg_2_Cl_2_ (s)|KCl (saturated)) and a Pt electrode were applied as a reference electrode and counter electrode, respectively. CV, DPV, and electrochemical impedance spectroscopy (EIS) tests were conducted to analyze the electrochemical behavior of different concentrations of DA and UA on the GCE modified with various materials. CV curves (five cycles) were recorded from 0 to 0.5 V with scan rate of 0.1 V/s, while DPV tests were conducted from −0.2 to 0.5 V with an increment step of 4 mV, amplitude of 50 mV, and pulse period of 0.5 s. EIS was performed in 0.1 to 100 KHz on various modified electrodes with 10 mV amplitude of the AC voltage.

## 3. Results and Discussion

### 3.1. Characterization of Ti_3_C_2_T_x_/rGO Nanocomposite

The schematic diagram of the simultaneous electrochemical detection procedures of DA and UA on Ti_3_C_2_T_x_/rGO electrode is displayed in Figure 1a. The uniform Ti_3_C_2_T_x_ and GO water dispersion were prepared through ultrasonication. GO dispersion was dropped and dried on the Ti_3_C_2_T_x_ electrode to form a Ti_3_C_2_T_x_/GO electrode with a double-deck structure, and an electrochemical reduction process was applied using CV sweeping to obtain a Ti_3_C_2_T_x_/rGO electrode for DA and UA detection. Based on previous studies (Figure 1b) [41], a pair of reversible peaks (Ox_1_, Re_1_) can be interpreted as the two-electron oxidation of DA to o-dopaminoquinone. Meanwhile, a pair of reversible peaks (Ox_2_, Re_2_) originated from the transformation of UA to dehydrourate. Specifically, oxidation peaks Ox_1_, Ox_2_ at 0.185 V and 0.316 V were chosen as the characteristic peaks for quantitative analysis of the electrochemical behavior of DA and UA, respectively.

According to the morphologies of Ti_3_C_2_T_x_ shown in Figure 2a, Ti_3_C_2_T_x_ was well distributed on the surface of GCE, and a porous electrode with good electrical conductivity was formed. Figure 2b is an enlarged view of Figure 2a, and the corresponding EDS mapping of C, Ti, F, and O are shown. The results indicate the two-dimensional layered sheet-like structures of Ti_3_C_2_T_x_ with good flatness. Figure 2c displays the morphologies of Ti_3_C_2_T_x_/rGO, exhibiting the rough surface of rGO with random wrinkles and the layered structures of Ti_3_C_2_T_x_. Figure 2d is an enlarged view of Figure 2c, and the corresponding EDS mapping of C, Ti, F, and O are shown. The morphology revealed the recovery of rGO film to the surface of Ti_3_C_2_T_x_.

The Raman spectra of rGO, Ti_3_C_2_T_x_ and Ti_3_C_2_T_x_/rGO are shown in Figure 2e. Three main peaks of rGO, namely the D band (~1350 cm^−1^), G band (~1580 cm^−1^), and 2D band (~2700 cm^−1^), correspond to random vibration of amorphous carbon (sp3 hybrid carbon) and in-plane vibration of graphitic carbon (sp2 hybrid carbon) [42]. The peak positions of Ti_3_C_2_T_x_ at 199 and 719 cm^−1^ are assigned to the out-of-plane vibrations of Ti and C atoms. The modes at 287, 369, and 624 cm^−1^ are the E_g_ group vibrations, including in-plane modes of Ti and C, and surface functional group atoms [43]. The peak positions of Ti_3_C_2_T_x_/rGO are located at 205, 287, 369, 624, 723, 1350, 1580, and 2700 cm^−1^, verifying the existence of both Ti_3_C_2_T_x_ and rGO in the composite. From the full XPS survey spectra of Ti_3_C_2_T_x_/rGO, F 1s, Ti 2s, O 1s and Ti 2p, C 1s appear at the binding energy of 684.8, 536.6, 496.9, 462.8 and 287.7 eV, respectively, as shown in Figure 2f, confirming the presence of four elements in the composite [44]. As depicted in Figure 2g, the Ti 2p narrow spectra of Ti_3_C_2_T_x_/rGO are divided into two parts: Ti 2p_3/2_ and Ti 2p_1/2_. Ti 2p_3/2_ spectra can be segmented into four components, which are located at 454.9 eV (Ti-C), 455.4 eV (Ti(II)), 456.3 eV (Ti(III)), and 458.8 eV (TiO_2_). The Ti 2p_1/2_ spectra can be fitted into three components, which are located at 461.1 eV (Ti-C), 462.1 eV (Ti(II)), and 462.6 eV (Ti(III)). Next, C 1s’ XPS curve can be fitted into five components (Figure 2h), which correspond to 284.8 eV (C-C), 281.6 eV (C-Ti), 282.4 eV (C-Ti-O), 287.5 eV (C=O), and 288.6 eV (O-C=O) [45]. Notably, the peak of C=O and O-C=O are ascribed to the introduction of rGO and the closed interaction between Ti_3_C_2_T_x_ and rGO [46]. The O 1s spectrum of Ti_3_C_2_T_x_/rGO is well fitted into two components (Figure 2i), which are centered at 531.8 eV (C-Ti-OH) and 529.6 eV (Ti-O-Ti). Therein, the oxygen-containing functional termination groups of Ti_3_C_2_T_x_/rGO were confirmed by the presence of C-Ti-OH bond [47]. The XPS consequence verifies the formation of Ti_3_C_2_T_x_/rGO heterostructure and is consistent with the previous results.

### 3.2. Electrochemical Collaboration Behavior of Ti_3_C_2_T_x_/rGO towards DA

To investigate the electrochemcial response of Ti_3_C_2_T_x_/rGO towards DA and UA, CV scanning was performed on the Ti_3_C_2_T_x_/rGO electrode in PBS with 10 μM DA and 10 μM UA. As shown in Figure 3a, compared with the CV curve from blank PBS, there is an oxidation peak and a reduction peak in the CV curve of DA (Re_1_, Ox_1_), and UA (Re_2_, Ox_2_) [41]. Among them, Ox_1_ and Ox_2_ were specified as the characteristic peaks for qualitative and quantitative analysis of the electrochemical behavior of DA and UA, respectively. As shown in Figure 3b,c, DPV curves of electrochemical behaviors at a potential interval of 0.0–0.5 V were conducted in the presence of 10 μM DA on bare GCE, Ti_3_C_2_T_x_, rGO, and Ti_3_C_2_T_x_/rGO electrodes. The current intensity of the Ti_3_C_2_T_x_ electrode exhibited higher than GCE, indicating that the porous Ti_3_C_2_T_x_ electrode with good electrical conductivity promoted the electron transfer of DA oxidation. Compared to the Ti_3_C_2_T_x_ electrode, the current intensity of the rGO electrode improved by nearly double, demonstrating the much better electrochemical performance of rGO than Ti_3_C_2_T_x_ towards DA. Furthermore, the Ox_1_ current intensity of Ti_3_C_2_T_x_/rGO electrode was much higher than the sum of rGO and Ti_3_C_2_T_x_, owing to the synergistic effect of the huge specific surface area of rGO and the porous Ti_3_C_2_T_x_ electrode with good electrical conductivity. To assess the electrochemcial feasibility of various modified electrodes in 10 mM [Fe(CN)_6_]^3−/4−^, EIS was performed on bare GCE, Ti_3_C_2_T_x_, and Ti_3_C_2_T_x_/rGO electrodes with 10 mV amplitude of the AC voltage, as shown in Figure 3d. The semicircle diameter at higher frequencies in the Nyquist diagram indicates the interfacial electron transfer resistance (R_ct_), which controls the electron transfer of [Fe(CN)_6_]^3−/4−^ on the electrode surface [48]. The R_ct_ values of GCE, Ti_3_C_2_T_x_, and Ti_3_C_2_T_x_/rGO electrodes were 1036.0, 628.8, and 369.6 Ω, respectively. The result reveals that the Ti_3_C_2_T_x_/rGO electrode greatly facilitates the electron transfer of the DA electrochemical reaction, which agrees with the former results. R_s_, R_p_, Q_coat_, and Q_sub_ represent the solution resistance, pore resistance, coating constant phase, and double-layer constant phase, respectively, and the corresponding values are listed in Appendix A.

To further investigate the synergistic effect of the Ti_3_C_2_T_x_/rGO nanocomposite, GO and Nafion were taken as the coating layer hybrid with Ti_3_C_2_T_x_ and rGO as the coating layer hybridize with the Au electrode instead of the Ti_3_C_2_T_x_ electrode, in comparison with the Ti_3_C_2_T_x_/rGO nanocomposite modified electrode. As shown in Figure 3e,f, the current intensity on the Ti_3_C_2_T_x_/Nafion electrode was lower than that of the Ti_3_C_2_T_x_ electrode. This indicates that Nafion as an electric material is not suitable for hybridizing with Ti_3_C_2_T_x_ to DA electrochemical reaction. The current intensity of the Ti_3_C_2_T_x_/rGO electrode was higher than of the Ti_3_C_2_T_x_/GO electrode, indicating that less oxygen-containing groups of rGO with better electrical conductivity exhibited greater facilitation of electron transfer reaction of DA. Interestingly, the current intensity of Au electrode/rGO electrode was lower than that of the Ti_3_C_2_T_x_/rGO electrode, demonstrating the advantage of the porous Ti_3_C_2_T_x_ electrode to the smooth Au electrode towards DA electrochemical reaction, as displayed in Appendix A. To further reveal the superior electrochemical performance of rGO to Ti_3_C_2_T_x_, a comparison experiment of DA adsorption performance was conducted between Ti_3_C_2_T_x_ and rGO water dispersions, as shown in Figure 3g. Ti_3_C_2_T_x_ and rGO water dispersions containing 100 μM DA were prepared with sonification. After filtration by 0.22 μM membrane, the filtrates of Ti_3_C_2_T_x_ and rGO dispersions both became much more transparent, and the colors of both film membranes were much darker. The result demonstrates that nanomaterials such as Ti_3_C_2_T_x_ and rGO adsorption DA were mostly trapped on the membrane. Then, the DPV curves of electrochemical behaviors at a potential interval of 0.0–0.35 V were conducted on the Ti_3_C_2_T_x_/rGO electrode in electrolytes using original DA solutions, filtrates of the Ti_3_C_2_T_x_, and rGO dispersions containing 300 μM DA, respectively, as shown in Figure 3h. The adsorption consequence revealed that the DA adsorption performance of rGO was greater than that of Ti_3_C_2_T_x_. This may be due to the electrostatic interaction between positively charged DA (pKa = 8.87) and negatively charged rGO with oxygen-containing groups at pH 7.0, as well as the π–π interaction between the phenyl structure of DA and two-dimensional planar hexagonal carbon–carbon structure of graphene, rather than the electrostatic interaction between DA and negative Ti_3_C_2_T_x_ only [49].

### 3.3. Ti_3_C_2_T_x_/rGO Electrode Performance Optimization of DA Detection

To further improve the electrochemical performance of the proposed sensor, experimental parameters including the preparation of modified electrodes, electrolyte pH were optimized. To affirm the influence of the preparation of layer-by-layer structured Ti_3_C_2_T_x_/rGO electrode on the DPV response in PBS with 100 nM DA, various rGO masses including 0.03, 0.075, 0.15, 0.3, 0.75, and 1.5 μg were formed on the same Ti_3_C_2_T_x_ electrode cast on 6.0 μg. As shown in Figure 4a, the background current of the DA detection peak in 0.124 V greatly increased with the rising mass of rGO, and the DA oxidation peak was obviously observed only when rGO mass adjusted to 0.075 μg in the fabrication of the Ti_3_C_2_T_x_/rGO electrode. The results reveal that the rising mass of rGO was not suitable for the nM concentration level of effective DA detection, and rGO mass was chosen as 0.075 μg. Similarly, to confirm the suitable mass of Ti_3_C_2_T_x_ in the fabrication of Ti_3_C_2_T_x_/rGO electrode, various Ti_3_C_2_T_x_ masses of 0.6, 1.5, 3, 6, 12, and 30 μg were firstly cast on GCE, and 0.075 μg GO was then dropped on and dried to perform the CV method of electrochemical reduction. The DPV response in PBS with 10 μM DA was then performed, as shown in Figure 4b,c. The Ti_3_C_2_T_x_ mass was selected as 3 μg, and the corresponding optimal mass ratio of Ti_3_C_2_T_x_ to rGO was 40:1.

To better determine if the mechanism of a lower rGO mass is suitable for trace level DA detection, CV and DPV curves at a potential interval of 0.0–0.5 V were performed in PBS on an rGO electrode cast with masses of 0.15 μg, 0.6 μg, and 3.0 μg respectively, as shown in Figure 4d,e. The capacitance can be calculated by CV methods with the following formula: C = $\frac{\int_{V0}^{V0+\Delta V}i\;dV}{S\cdot \Delta V}$, wherein, *S* refers to scan rate; Δ*V* refers to potential scan range; and *i* refers to current. The area A_c_ of the CV curves determines the value of capacitance, when *S* and Δ*V* remain consistent. Obviously, the results reveal that the total capacitance formed on the rGO electrode and the background current both increased with the rising rGO mass. The structure of the EDL at the junction of a metal with an electrolyte solution conceives the layer to have two elements, known as “Helmholtz layer, and diffuse layer” [50]. The two elements interpret the existence of a capacitance C_d_ of electrical double layer to be close to the solid/electrolyte interface, the Helmholtz capacitance C_H_, and diffuse layer capacitance C_D_, wherein C_d_^−1^ = C_H_^−1^ + C_D_^−1^ [51]. The thickness of the diffuse layer gives the distance from the solution up to the point where the electrostatic effect of the surface is felt by the ions [40]. According to the schematic diagram in Figure 4f,g, when the rGO mass modified on GCE increased, the total capacitance C_d_ increased, and the diffuse layer capacitance C_D_ increased. Thus, λ_D_ decreased with respect to the rGO electrode modified with greater rGO mass, indicating that fewer DA biomolecules were capable of passing through EDL and reaching the surface of GO to achieve the voltammetric response of DA. The increasing mass of GO decreased λ_D_, suggesting that the detection limit of DA was raised to a higher level, and the result is consistent with Figure 4a.

The effect of pH on the electrochemical response of the Ti_3_C_2_T_x_/rGO electrode was conducted in the range from 3.0 to 11.0, as shown in Figure 4h. The oxidation peak potentials of DA shifted negatively with the increased electrolyte pH, ascribing to an improvement in the reversibility of the investigated faradic process that involves the deprotonation of DA, followed by the protonation of the amine group in DA to form a cation [52,53]. The value of the peak current reached the maximum at pH 7.0 and was selected as the optimal pH value. The electrochemical behavior of various electrodes was performed by CV in 10 mM [Fe(CN)_6_]^3−/4−^ containing 0.1 M KCl electrolyte solution at scan rates ranging from 20 to 260 mV s^−1^ (Figure 4i). The observed peak currents (I_pa_ and I_pc_) both increased linearly, with the square root of scan rates as shown in Figure 4j, indicating that the Ti_3_C_2_T_x_/rGO electrodes were controlled by diffusion [54].

### 3.4. Electrochemical Determination of DA and UA with Different Concentrations

The quantitative electrochemical detection of DA and UA on the Ti_3_C_2_T_x_/rGO electrode was conducted via DPV measurements, as shown in Figure 5a,b. An increase in peak current value was recorded with the increasing concentration of DA in a range from 9.5 nM to 100 μM, and the increasing concentration of UA in a range from 300 nM to 1000 μM, respectively. Correspondingly, the inset graphic of Figure 5a,b depicts the enlarged view in the potential range from 0.0–0.3 V and 0.1–0.4 V to clearly show the variations of DPV curves ranging from 0.0–100 nM DA and 0.0–1.0 μM DA, respectively. As presented in Figure 5c, the calibration curve of DA and UA was obtained from the average of peak current data. According to the calibration curve, the linear range of DA detection was in a range from 0.1 to 100 μM, and UA detection was in a range from 1 to 1000 μM, respectively. The linear regression equation of DA was I_pc_ (μA) = 0.413 lg DA (μM) − 5.780 (R^2^ = 0.993), and the linear regression equation of UA was I_pc_ (μA) = 0.529 lg UA (μM) − 0.209 (R^2^ = 0.994). The limit of detection (LOD) of DA and UA on the Ti_3_C_2_T_x_/rGO electrode was determined as 9.5 nM and 300 nM, respectively.

Specifically, the quantitative electrochemical detection of DA and UA were conducted by DPV measurements on six individual electrodes. Compared with graphene- based modified electrodes prepared using various methods for simultaneous detection of DA and UA, as shown in Figure 5d, our Ti_3_C_2_T_x_/rGO electrode achieved a relatively low simultaneous detection LOD of DA and UA and a four-order-magnitude linear range with convenience and efficiency. The corresponding literatures are listed in Table 1.

### 3.5. Repeatability, Reproducibility, Interference, and Real Sample Analysis

In order to study the repeatability of the Ti_3_C_2_T_x_/rGO electrode for DA detection (10 μM), the Ox_1_ peak currents in DPV curves were repeatedly measured 11 times on the same electrode at a potential interval of 0.0–0.5 V. As shown in Figure 5e, the reduction peak potentials of DPV curves were consistent at 0.129 V, and these curves overlapped well. The relative standard deviation (RSD) of peak currents was 3.57%. The reproducibility of the Ti_3_C_2_T_x_/rGO electrode was performed in the presence of 10 μM DA by using six individual electrodes in DPV curves, as shown in Figure 5f, and the RSD was 3.92%. The results indicate that the Ti_3_C_2_T_x_/rGO electrode has good repeatability and reproducibility. The anti-interference of the Ti_3_C_2_T_x_/rGO electrode was investigated via DPV curves in PBS containing various concentrations of DA ranging from 0.1 μM to 10 μM in the presence of 30 μM UA as interfering substances, as shown in Figure 5g. Similarly, the anti-interference was performed in PBS containing UA ranging from 1 μM to 100 μM in the presence of 10 μM DA, as presented in Figure 5h. When compared to the calibration curve of DA and UA detection individually in Figure 5a,b, the anti-interference results indicate that DA and UA did not induce obvious interference in the DPV determination of each other. The anti-interference of Ti_3_C_2_T_x_/rGO electrode in the presence of other potential interfering substances as 100 µM glucose, 100 µM ascorbic acid (AA), 100 µM H_2_O_2_, and 10 µM isoniazid in PBS containing 3 μM DA and 3 μM UA was investigated via DPV curves, as shown in Appendix A. The results indicate that our constructed sensor will not be affected by these molecules during testing.

To evaluate the practical application performance of Ti_3_C_2_T_x_/rGO electrodes for simultaneous detection of DA and UA, human serum was selected as real samples for analysis using the standard addition technique. The serum samples were centrifuged at 6000 rpm for 5 min, and the supernatants were collected and diluted 100 times with PBS. Then KCl was added to 0.1 M of the serum samples, and the pH was adjusted to 7.0 to perform appropriate electrochemical detection of DA and UA [56]. Serum samples were then spiked with 0.1, 0.3 μM DA, and 1, 3 μM UA, respectively, and the DPV curves of Ti_3_C_2_T_x_/rGO electrode were extracted, as shown in Figure 5i. The results demonstrate the accuracy and reliability of the fabricated sensor, indicating that the proposed Ti_3_C_2_T_x_/rGO electrode exhibited good potential for simultaneously detecting DA and UA practically.

Thus, our fabricated Ti_3_C_2_T_x_/rGO electrode with a double-deck layer was applied as the biosensor for the simultaneous detection of DA and UA successfully. The detection sensitivity of the Ti_3_C_2_T_x_/rGO electrode was greatly improved with the adjustment to Debye length. Our proposed Ti_3_C_2_T_x_/rGO electrode displayed good repeatability, selectivity, and proved suitable for real sample analysis.

## 4. Conclusions

In summary, a Ti_3_C_2_T_x_/rGO heterostructure with a double-deck layer was fabricated through electrochemical reduction. The rGO was modified on the porous Ti_3_C_2_T_x_ electrode as the biosensor for the simultaneous detection of DA and UA. The Debye length λ_D_ is regulated by the alteration of rGO on the surface of the Ti_3_C_2_T_x_ electrode. λ_D_ decreased with respect to the rGO electrode modified with a greater rGO mass, indicating that fewer DA biomolecules were capable of passing through EDL and reaching the surface of GO to achieve the voltammetric response of DA. Thus, the proposed Ti_3_C_2_T_x_/rGO sensor had an excellent performance in the detection of DA and UA, with a wide linear range from 0.1–100 μM to 1–1000 μM and a low detection limit from 0.0095 to 0.3 μM, respectively. Additionally, the proposed Ti_3_C_2_T_x_/rGO electrode displayed good repeatability, selectivity, and proved suitable for real sample analysis.

## Figures and Tables

**Figure 1 biosensors-11-00462-f001:**
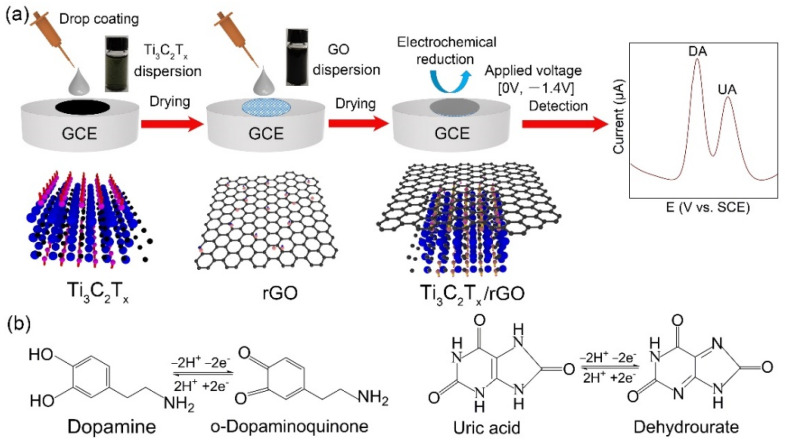
(**a**) Schematic diagram of electrochemical detection of DA and UA based on Ti_3_C_2_T_x_/rGO electrode. (**b**) The proposed reaction scheme of redox reaction of DA and UA during electrochemical detection, respectively.

**Figure 2 biosensors-11-00462-f002:**
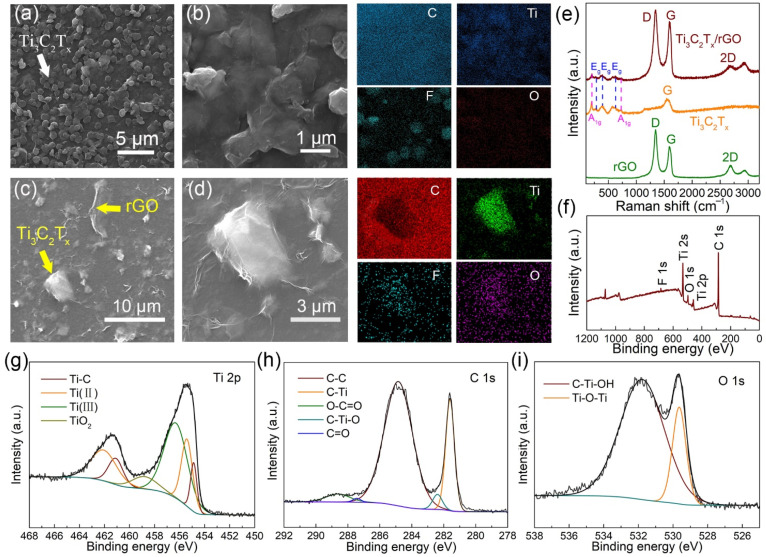
SEM images of Ti_3_C_2_T_x_ electrode (**a**) and Ti_3_C_2_T_x_/rGO electrode (**c**). (**b**,**d**) Regional enlarged view of (**a**,**c**) and the EDS mapping of element distribution of C, Ti, F, O, respectively. (**e**) Raman spectra of Ti_3_C_2_T_x_, rGO, and Ti_3_C_2_T_x_/rGO nanocomposite. (**f**) XPS survey spectra of Ti_3_C_2_T_x_/rGO, and Ti 2p spectra (**g**), C 1s spectra (**h**), O 1s spectra (**i**) spectra, respectively.

**Figure 3 biosensors-11-00462-f003:**
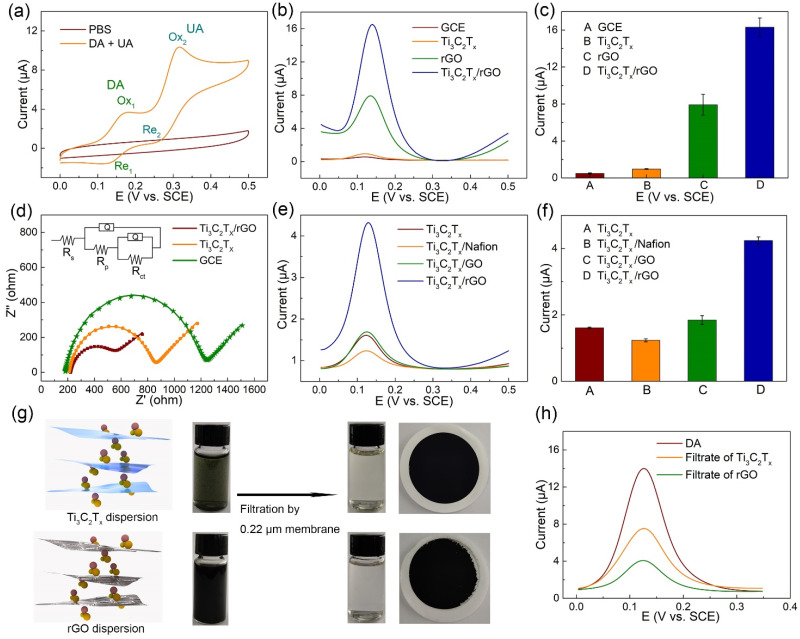
(**a**) CV of Ti_3_C_2_T_x_/rGO electrode with and without 10 μM DA and 30 μM UA in PBS. (**b**) DPV curves of various modified electrodes with 10 μM DA in PBS. (**c**) The corresponding current value of (**b**). (**d**) Impedance plots of various modified electrodes with 10 mM [Fe(CN)_6_]^3−/4−^. (**e**) Performance comparison of DPV curves on various materials modified Ti_3_C_2_T_x_ electrode with 10 μM DA in PBS. (**f**) The corresponding current value of (**e**). (**g**) Schematic diagram of DA adsorption in Ti_3_C_2_T_x_ and rGO dispersion. (**h**) DPV curves of DA adsorption performance.

**Figure 4 biosensors-11-00462-f004:**
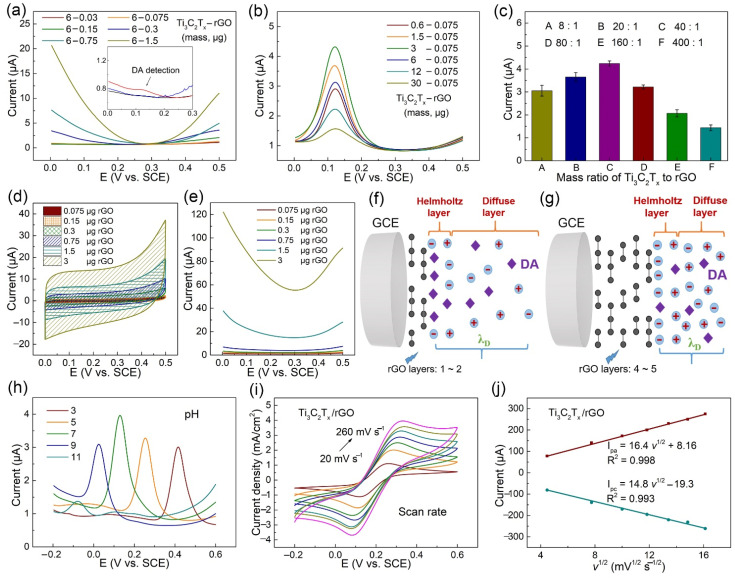
(**a**) DPV curves of Ti_3_C_2_T_x_/rGO electrode with various masses of rGO to the same of Ti_3_C_2_T_x_ with 100 nM DA in PBS; the inset graphic depicts an enlarged view in the potential range between 0.0–0.3 V. (**b**) DPV of Ti_3_C_2_T_x_/rGO electrode with various masses of Ti_3_C_2_T_x_ to the same of rGO with 10 μM DA in PBS. (**c**) the corresponding current value of (**b**). (**d**,**e**) CV and DPV of rGO electrode in PBS with various amounts of rGO, respectively. (**f**,**g**) Schematic diagram of EDL model to interpret DA detection mechanism via Debye length regulation. (**h**) DPV of 10 μM DA on Ti_3_C_2_T_x_/rGO electrode with pH. (**i**) Ti_3_C_2_T_x_/rGO electrode in 10 mM [Fe(CN)_6_]^3−/4−^ and 0.1 M KCl electrolyte solution at scan rates (*v*) from 20 to 260 mV s^−1^. (**j**) Linear plots of I_pa_/I_pc_ vs. *v*.

**Figure 5 biosensors-11-00462-f005:**
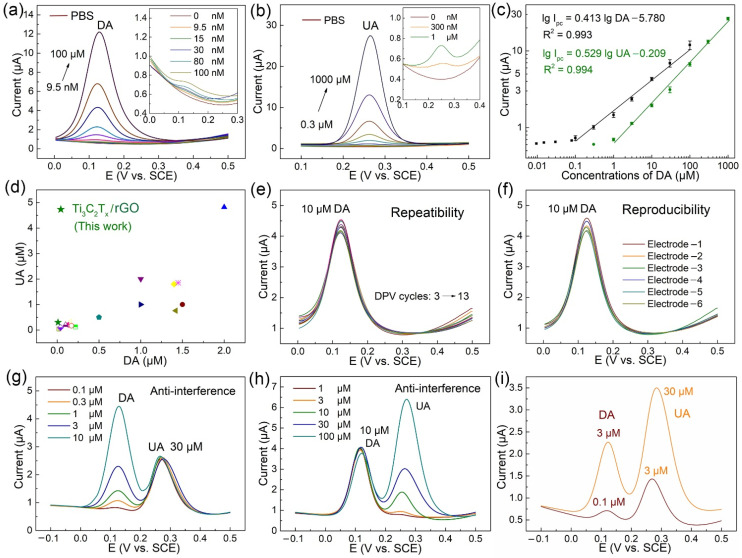
(**a**,**b**) DPV curves of Ti_3_C_2_T_x_/rGO electrode with various concentrations of DA and UA; the inset graphic depicts an enlarged view in the potential range between 0.0–0.3 V and 0.1–0.4 V, respectively. (**c**) The corresponding peak current versus DA and UA concentration. (**d**) The performance comparisons. (**e**,**f**) The repeatability and reproducibility of Ti_3_C_2_T_x_/rGO electrode. (**g**,**h**) Good anti-interference of our electrodes. (**i**) Serum sample analysis.

**Table 1 biosensors-11-00462-t001:** Performance comparison of graphene-based materials modified electrodes for simultaneous detection of DA and UA.

Modified Electrodes	Measurements	Linear Range (μM)		LOD (μM)		Ref.
		DA	UA	DA	UA	
rGO	DPV	0.5–60	0.5–60	0.5	0.5	[49]
Graphene	Amperometric	5.0–710	6.0–1330	2.0	4.8	[24]
Graphene	DPV	0.5–2000	0.8–2500	0.12	0.2	[14]
Graphene aerogel	DPV	0.65–75	0.4–50	0.22	0.12	[17]
CNTs/GO	DPV	5.0–500	3.0–60	1.5	1.0	[29]
Chitosan/Graphene	DPV	1.0–24	2.0–45	1.0	2.0	[55]
Au/rGO	DPV	6.8–41	8.8–53	1.4	1.8	[27]
Au/Pt/GO/rGO	DPV	0.07–49,800	0.13–82,800	0.02	0.04	[56]
Pt NPs/Graphene	DPV	0.03–8.13	0.05–11.9	0.03	0.05	[28]
Ag/rGO	DPV	10–70	10–130	1.0	1.0	[57]
Pd/Pt/rGO	DPV	4–200	4–400	0.04	0.1	[6]
Mn_3_O_4_/rGO	SWV ^a^	1–600	1–600	1.42	0.76	[58]
Hemin/GO	DPV	0.5–40	0.5–50	0.17	0.17	[59]
TiN/rGO	DPV	5–175	30–215	0.16	0.35	[60]
N-doped rGO	DPV	1–60	1–30	0.1	0.2	[61]
Ti_3_C_2_T_x_/rGO	DPV	0.1–100	1–1000	0.0095	0.3	This work

^a^ SWV: Square wave voltammetry.

## Data Availability

Data sharing not applicable.

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
