# Peer review of "A Double-Deck Structure of Reduced Graphene Oxide Modified Porous Ti3C2Tx Electrode towards Ultrasensitive and Simultaneous Detection of Dopamine and Uric Acid"

_biosensors, 2021, doi:10.3390/bios11110462_

Round 1

Reviewer 1 Report

Zhu et al. report on the “A double-deck structure of rGO modified porous Ti3C2Tx electrode towards ultrasensitive and simultaneous detection of DA and UA.” The content of the work is interesting, but the manuscript cannot be published in the present form due to the following issues:

  1. XRD of Ti3C2Tx is missing.
  2. Cross-sectional FESEM images of Ti3C2Tx/ rGO is also required.
  3. Please mention the values of the different entities such as Rs, Rp, Rct for all the curves plotted in a tabular form also add the values of Q’s. What is the potential which has been kept in acquiring the EIS data?
  4. Why the rGO mass which was cast has not been taken in particular proportional intervals as it is taken as 0.15 µg, 0.6 µg, and directly 3 µg. what happens between the interval 0.6µg-3 µg. Also, the authors should show variations lower than 0.15µg.
  5. Explain the physical significance of humps shown at the inset of both Figure 5 (a) and (b) at nM levels
  6. Please precisely mention the outcomes regarding work presented in brief as a state of art before the conclusion.

Author Response

Dear Editor,

We are very grateful to your and the reviewers’ critical comments and thoughtful suggestions. Based on these comments, we have made a careful revision of the original manuscript (Manuscript ID: biosensors-1437418). A revised manuscript has been resubmitted, of which the modified sections are marked in red. Thank you and reviewers again, who made great contribution to improve our paper. We responded point by point to the reviewers’ comments as listed below, along with a clear indication of the location in the manuscript:

Reviewer #1:

  1. XRD of Ti3C2Txis missing.

Reply: Thank you for your suggestion. XRD of Ti3C2Tx is provided, as shown in Figure R1. The (002), (004), and (110) peaks are characteristic to delaminated Ti3C2Tx [1].

Figure R1. XRD pattern of Ti3C2Tx.

  1. Cross-sectional FESEM images of Ti3C2Tx/ rGO is also required.

Reply: Thank you for your suggestion. Cross-sectional and frontal FESEM images of Ti3C2Tx/ rGO are provided, as shown in Figure R2 and R3.

Figure R2 (a). Cross-sectional FESEM images of Ti3C2Tx/ rGO. (b) the corresponding linear scanning of (a).

Figure R3. Double-deck structure SEM image of Ti3C2Tx/ rGO.

  1. Please mention the values of the different entities such as Rs, Rp, Rct for all the curves plotted in a tabular form also add the values of Q’s. What is the potential which has been kept in acquiring the EIS data?

Reply: Thank you for your suggestion. Rs, Rp, Qcoat, Rct and Qsub represent the solution resistance, pore resistance, coating constant phase, charge transfer resistance and double-layer constant phase, respectively. According to the fitting curves in addition to the experimental data points in Figure 3d, the values of the different entities as Rs, Rp, Qcoat, Rct and Qsub are listed respectively in the Table S1. The potential kept in acquiring the EIS data for GCE, Ti3C2Tx and Ti3C2Tx/rGO electrode is 0.196, 0.194, 0.194 V, respectively.

   Table S1. The fitting parameters of EIS for GCE, Ti3C2Tx and Ti3C2Tx/rGO electrode.

Electrode

Rs (Ω)

Qcoat (F)

Rp (Ω)

Qsub (F)

Rct (Ω)

GCE

181.1

1.49×10-6

2052

2.84×10-3

1036.0

Ti3C2Tx

210.0

1.68×10-6

2209

2.52×10-3

628.8

Ti3C2Tx/rGO

221.4

1.50×10-4

596.4

3.83×10-3

369.6

  1. Why the rGO mass which was cast has not been taken in particular proportional intervals as it is taken as 0.15 µg, 0.6 µg, and directly 3 µg. what happens between the interval 0.6µg-3 µg. Also, the authors should show variations lower than 0.15µg.

Reply: Thank you for your suggestion. CV and DPV of rGO electrode in PBS have been provided with various amounts of rGO ranging from 0.075 to 3 µg respectively, to keep consistent with the dropping mass of GO in the preparation of Ti3C2Tx/rGO electrode in the manuscript of Figure 4a . The CV and DPV curves are shown in Figure R4.

Figure 4a.

Figure R4 (a, b). CV and DPV of rGO electrode in PBS with various amounts of rGO, respectively.

  1. Explain the physical significance of humps shown at the inset of both Figure 5 (a) and (b) at nM levels.

Reply: Thank you for raising this question. The inset graphic of Figure 5a and Figure 5b is to clearly observe the variations of DPV curves ranging from 0 nM – 9.5 nM DA and 0 nM –1 μM DA, respectively. These DPV curves at nM levels show the LOD of DA and UA, which were 9.5 nM, and 300 nM, respectively. The result indicates that our Ti3C2Tx/rGO electrode achieved a relatively low simultaneous detection LOD of DA and UA and a four-order-magnitude linear range with convenience and efficiency. A description about this conclusion has been added in the manuscript in Section 3.4 in Results and Discussion (Paragraph 1, Line 13-14; Paragraph 2, Line 2-6).

Figure 5 a and b.

  1. Please precisely mention the outcomes regarding work presented in brief as a state of art before the conclusion.

Reply: Thank you for your suggestion. A precisely mention in brief has been provided, as “Thus, our fabricated Ti3C2Tx/rGO electrode with double-deck layer was applied as the biosensor to the simultaneous detection of DA and UA successfully. The detection sensitivity of Ti3C2Tx/rGO electrode was improved much with the adjustment of Debye length. Our proposed Ti3C2Tx/rGO electrode displayed good repeatability, selectivity, and proved suitable for real sample analysis.” A description about this conclusion has been added in the manuscript in Section 3.5 in Results and Discussion (Paragraph 3).

References:

[1] Rasheed, P.A.; Pandey, R.P.; Rasool, K.; Mahmoud, K.A. Ultra-sensitive electrocatalytic detection of bromate in drinking water based on Nafion/Ti3C2Tx (MXene) modified glassy carbon electrode. Sens. Actuators, B: Chem. 2018, 265, 652–659.

We appreciate for Editor/Reviewers’ warm work earnestly, and hope that the correction will meet with approval. The manuscript has been overall checked, and the changes marked in red font one by one. We hope that these revisions are sufficient to make our manuscript acceptable for publication in Biosensors. If you have any question about this paper, please do not hesitate to contact me.

Yours sincerely,

Cheng-Te Lin

Ningbo Institute of Material Technology & Engineering, Chinese Academy of Sciences

Reviewer 2 Report

Dear authors,

1) The manuscript needs an extensive editing of English language and style.

2)  The developed sensors are not biosensors since no biological receptor for analyte selective recognition is involved accordingly to the biosensor definition

3) No details regarding the characteristics of:

  • GO powder and rGO water dispersion  purchased from Nanjing JCNANO Technology Co., Ltd.
  • Ti3C2Tx water dispersion purchased from Beike 2D materials Co., Ltd.

were provided for the assessment of the further experiments.

4) The proposed mechanism of the DA and UA electrochemical oxidation is not experimental supported.

5) The effect of pH cannot be done in PBS since PBS has not buffering capcity at pH 3 or pH 11.

6) No interference study against other potential interfering compound (i.e.  ascorbic acid, peroxide radicals, etc) was done.

7) Many other small, but important issues must be considered in a deeper way.

Author Response

Dear Editor,

We are very grateful to your and the reviewers’ critical comments and thoughtful suggestions. Based on these comments, we have made a careful revision of the original manuscript (Manuscript ID: biosensors-1437418). A revised manuscript has been resubmitted, of which the modified sections are marked in red. Thank you and reviewers again, who made great contribution to improve our paper. We responded point by point to the reviewers’ comments as listed below, along with a clear indication of the location in the manuscript:

Reviewer #2:

  1. The manuscript needs an extensive editing of English language and style.

Reply: Thank you for your suggestions. The English language and style has been extensively edited.

  1. The developed sensors are not biosensors since no biological receptor for analyte selective recognition is involved accordingly to the biosensor definition.
  • Reply: Thank you for raising this question. A biosensor can be defined as a self-contained analytical device that combines a biological component with a physicochemical component for the detection of an analyte of biological importance [1]. Biological component comprises nucleic acids, proteins including enzymes and antibodies tissue slices, microorganisms and organelles in a narrow sense. Biosensors can be of immense importance in tissue engineering applications, particularly concerning to that biomolecules such as glucose, adenosines, dopamine, and hydrogen peroxide levels play important roles in determining the fate of the cells and tissues [2]. Living cells are well known to transmit various physical and chemical signals, such as changes in consumption of oxygen, pH, membrane potentials, ion concentrations, and release of various metabolic compounds and proteins [3]. Monitoring these analytes can give insights into cellular activities in real time. Recently, nonenzymatic detection of biomolecules using graphene-based electrodes to constructed electrochemical biosensors have attracted significant attention. For example, Zhong et al., showed the effect of AgNPs on the graphene thin films for the nonenzymatic detection of H2O2 [4]. Yuan et al., constructed the graphene–diamond hybrid electrode as electrochemical biosensors to detect of dopamine in nonenzymatic detection method [5]. Our work is to construct Ti3C2Tx/rGO electrode to nonenzymatic detect biomolecules as DA and UA, and thus can be regarded as electrochemical biosensors in a broad sense.
  1. No details regarding the characteristics of:
  • GO powder and rGO water dispersion purchased from Nanjing JCNANO Technology Co., Ltd.
  • Ti3C2Tx water dispersion purchased from Beike 2D materials Co., Ltd. were provided for the assessment of the further experiments.
  • Reply: We acknowledge your Figure R1 (a, b) provides the photograph information of GO powder and GO water dispersion (c) purchased from Nanjing JCNANO Technology Co., Ltd. Figure R1 (d) depicts the Raman spectra of GO and rGO, and proves the existence of these molecular structures. Figure R2 (a, b) provide the photograph information of Ti3C2Tx water dispersion used in the experiments, and Figure R2 (c) proves the elemental composition of Ti3C2Tx.

Figure R1. The photographs of GO power (a, b) and rGO dispersion (c), respectively. (d) Raman spectra of GO and rGO.

Figure R2. The Photographs of Ti3C2Tx water dispersion (a, b). Raman spectra of Ti3C2Tx (c).

  1. The proposed mechanism of the DA and UA electrochemical oxidation is not experimental supported.

Reply: Thank you for your suggestion. According to the Figure 1b in the manuscript, a pair of reversible peaks (Ox1, Re1) can be interpreted as the two electron oxidation of DA to

o-Dopaminoquinone, and a pair of reversible peaks (Ox2, Re2) were originated from the transformation of UA to Dehydrourate. The proposed mechanism of the DA and UA electrochemical oxidation is consistent with the previous literatures as [6]. Furthermore, as shown in Figure R3 (a), there are a pair of oxidation peak and reduction peak in the CV curve of DA (Re1, Ox1), and UA (Re2, Ox2) separately. As shown in Figure R3 (b), compared with the CV curve from blank PBS, there are both a pair of oxidation peak and reduction peak in the CV curve of DA (Re1, Ox1), and UA (Re2, Ox2). These experimental data all support the proposed mechanism of the DA and UA electrochemical oxidation well.

Figure R3. CV of Ti3C2Tx/rGO electrode with 10 μM DA or 30 μM UA in PBS separately (a), and with and without 10 μM DA and 30 μM UA (b).

  1. The effect of pH cannot be done in PBS since PBS has not buffering capcity at pH 3 or pH 11.

Reply: Thank you for rising this question. According to the other researching works reported [7, 8], the desired pH value ranging from 5 ~ 9 was also adjusted by HCl and NaOH. We acknowledge that PBS has no buffering capacity at pH 3 and 11, so we conduct the experiment by changing the buffer solution at pH 3 and 11. The buffer solution with pH 3 was adjusted by 4.11 mL of 0.2 M Na2HPO4, and 15.9 mL of citric acid, and the buffer solution with pH 11 was adjusted by adding 3g NH4Cl into 207 mL NH3.H2O to 500 mL volume. As shown in Figure R4, the electrochemical performance of Ti3C2Tx / rGO electrode with buffer solution replaced at pH 3 and 11 is not influenced.   

Figure R4. (a) DPV of 10 μM DA on Ti3C2Tx/rGO electrode with pH at 3 and 11. (b) the corresponding adjustment in the manuscript of Figure 4f.

  1. No interference study against other potential interfering compound (i.e. ascorbic acid, peroxide radicals, etc) was done.

Reply: Thank you for your suggestion. The anti-interference study of Ti3C2Tx/rGO electrode was investigated via DPV curves in PBS containing 3 µM DA and 3 µM UA in the presence of other potential interfering substances, such as 100 µM glucose, 100 µM ascorbic acid (AA), 100 µM H2O2 and 10 µM isoniazid, as shown in Figure R5. The oxidation peak of AA and isoniazid is appeared at potential of 0 V and 0.9 V, respectively, while H2O2 and AA is not appeared. The results suggests that these additive species did not induce obvious interference in simultaneous DPV determination of DA and UA. A description about this conclusion has been added in the manuscript in Section 3.5 in Results and Discussion (Paragraph 1, Line 16-20).

Figure R5. The anti-interference study.

  1. Many other small, but important issues must be considered in a deeper way.

Reply: Thank you for your suggestion. Important issues have been considered in a deeper way.

References:

[1] Hasan, A.; et al.; Recent Advances in Application of Biosensors in Tissue Engineering. Biomed. Res. In. 2014, 1-18.

[2] Wheeldon, I.; Farhadi, A.; Bick, A.G.; Jabbari, E.; and Khademhosseini, A. Nanoscale tissue engineering: spatial control over cell-materials interactions,” Nanotechnology, 2011, 22, 212001.

[3] Wang, Y.; Chen, Q.; and Zeng, X. Potentiometric biosensor for studying hydroquinone cytotoxicity in vitro. Biosens. Bioelectron, 2010, 25, 1356–1362.

[4] Zhong, L.J.; Gan, S.Y.; Fu, X.G.; Li, F.H.; Han, D.X.; Guo, L.P.; Niu, L. Electrochemically controlled growth of silver nanocrystals on graphene thin film and applications for efficient nonenzymatic H2O2 biosensor. Electrochim. Acta. 2013, 89, 222–228.

[5] Yuan, Q.L.; Liu, Y.; Ye, C.; Sun, H.Y.; Dai, D.; Wei, Q.P.; Lai, G.S.; Wu, T.Z.; Yu, A.M.; Fu, L.; Chee, K.W.A.; Lin, C.-T. Highly stable and regenerative graphene–diamond hybrid electrochemical
biosensor for fouling target dopamine detection. Biosens. Bioelectron, 2018, 111, 117–123.

[6] Yue, H.Y.; Huang, S.; Chang, J.; Heo, C.; Yao, F.; Adhikari, S.; Gunes, F.; Liu, L.C.; Lee, T.H.; Oh, E.S.; Li, B.; Zhang, J.J.; Huy, T.Q.; Luan, N.V.; Lee, Y.H. ZnO nanowire arrays on 3D hierachical graphene foam: biomarker detection of Parkinson's disease. ACS Nano. 2014, 8, 1639–1646.

[7] Feng, J.; Li, Q.; Cai, J.P.; Yang, T.; Chen, J.H.; Hou, X.M. Electrochemical detection mechanism of dopamine and uric acid on titanium nitride-reduced graphene oxide composite with and without ascorbic acid. Sens. Actuators, B: Chem. 2019, 298, 126872.

[8] Han, H.S; Lee, H.K.; You, J.-M.; Jeong, H.; Jeon, S. Electrochemical biosensor for simultaneous determination of dopamine and serotonin based on electrochemically reduced GO-porphyrin. Sens. Actuators, B: Chem. 2014, 190, 886­–895.

We appreciate for Editor/Reviewers’ warm work earnestly, and hope that the correction will meet with approval. The manuscript has been overall checked, and the changes marked in red font one by one. We hope that these revisions are sufficient to make our manuscript acceptable for publication in Biosensors. If you have any question about this paper, please do not hesitate to contact me.

Yours sincerely,

Cheng-Te Lin

Ningbo Institute of Material Technology & Engineering, Chinese Academy of Sciences

Reviewer 3 Report

Authors have prepared the reduced graphene oxide (rGO) - Ti3C2Tx  through electrochemical reduction and used it for the simultaneous electrochemical detection of dopamine (DA) and uric acid (UA). The rGO- Ti3C2Tx material synergy was increased the detection limit and sensitivity due to improved conductivity of rGO and increase in debye length.

  1. Figure 3d legend for EIS is not clear, fit and experimental data may be distinguished using line and symbol respectively
  2. Figure 3g legend has spelling mistake, ‘dispersion’
  3. Figure S2 (b) need to check the y axis scale, current value seems not comply with the DPV shown in Figure S2 (a).
  4. The authors could elaborate the would the order of optimization affect the loadings of Ti3C2Tx and rGO, start with rGO and Ti3C2Tx Why not Ti3C2Tx first and rGO after.
  5. On page 8, In Figure 4b and 4c discussion the optimized Ti3C2Tx mass is incorrect not match with the results seen on figure. The correct value as per the Figure 4b and c is 3 mg Ti3C2Tx. It may be a typo. Authors may need to go through the manuscript carefully to check the spelling and typo.
  6. Interference of the other common molecules (ascorbic acid) which oxidize in the same potential window may need to be included, to show the sensor won’t be affected by other molecules during testing.

Author Response

Dear Editor,

We are very grateful to your and the reviewers’ critical comments and thoughtful suggestions. Based on these comments, we have made a careful revision of the original manuscript (Manuscript ID: biosensors-1437418). A revised manuscript has been resubmitted, of which the modified sections are marked in red. Thank you and reviewers again, who made great contribution to improve our paper. We responded point by point to the reviewers’ comments as listed below, along with a clear indication of the location in the manuscript:

Reviewer #3:

  1. Figure 3d legend for EIS is not clear, fit and experimental data may be distinguished using line and symbol respectively.

Reply: Thank you for your suggestion. The fit and experimental data have been distinguished using line and symbol respectively in Figure 3d, as shown in the following:

Figure 3d

  1. Figure 3g legend has spelling mistake, ‘dispersion’.

Reply: Thank you for raising this question. The spelling mistake ‘dispertion’ has been corrected as ‘dispersion’ in Figure 3g.

  1. Figure S2 (b) need to check the y axis scale, current value seems not comply with the DPV shown in Figure S2 (a).

Reply: Thank you for your suggestion. The current value in Figure S2 (b) have been adjusted to comply with the DPV shown in Figure S2 (a).

Figure S2. (a) Performance comparison of DPV curves on Ti3C2Tx/rGO and Au/rGO electrode with 10 μM DA in PBS. (b) The corresponding current value of (a).

  1. The authors could elaborate the would the order of optimization affect the loadings of Ti3C2Txand rGO, start with rGO and Ti3C2Tx Why not Ti3C2Tx first and rGO after.

Reply: Thank you for raising this question. According to the Figure 4a, the oxidation peak of DA can’t be observed clearly when GO mass excessed 0.15 μg in the fabrication of Ti3C2Tx/rGO electrode. The result indicated that raising mass of rGO was not suitable for nM level DA detection effectively. If the amount of Ti3C2Tx is adjusted firstly while GO mass excessed 0.15 μg in the fabrication of Ti3C2Tx/rGO electrode, the oxidation peak of DA can’t be observed. The quantitative electrochemical detection of DA on Ti3C2Tx/rGO electrode can’t be conducted effectively via DPV measurements. Thus, rGO first and Ti3C2Tx after in the optimization adjustment of the loading amounts of Ti3C2Tx and rGO.

Figure 4a.

  1. On page 8, In Figure 4b and 4c discussion the optimized Ti3C2Txmass is incorrect not match with the results seen on figure. The correct value as per the Figure 4b and c is 3 mg Ti3C2Tx. It may be a typo. Authors may need to go through the manuscript carefully to check the spelling and typo.

Reply: Thank you for raising this question. The various concentrations of Ti3C2Tx was prepared as 0.1, 0.25, 0.5, 1, 2, 5 mg/mL, the dropping amount was 6 μL, and so various Ti3C2Tx mass was confirmed as 0.6, 1.5, 3, 6, 12, 30 μg, respectively. As shown in Figure 4b, the optimized Ti3C2Tx mass has been corrected as 3 μg, as mentioned in the manuscript in Section 3.3 (Line 15, Page 8). Other spelling and typos have been checked carefully throughout the manuscript.

Figure 4b.

  1. Interference of the other common molecules (ascorbic acid) which oxidize in the same potential window may need to be included, to show the sensor won’t be affected by other molecules during testing.

Reply: Thank you for your suggestion. The anti-interference study of Ti3C2Tx/rGO electrode was investigated via DPV curves in PBS containing 3 µM DA and 3 µM UA in the presence of other potential interfering substances, such as 100 µM glucose, 100 µM ascorbic acid (AA), 100 µM H2O2 and 10 µM isoniazid, as shown in Figure R1. The oxidation peak of AA and isoniazid is appeared at potential of 0 V and 0.9 V, respectively, while H2O2 and AA is not appeared. The results suggests that these additive species did not induce obvious interference in simultaneous DPV determination of DA and UA. A description about this conclusion has been added in the manuscript in Section 3.5 in Results and Discussion (Paragraph 1, Line 16-20).

Figure R1. The anti-interference study.

We appreciate for Editor/Reviewers’ warm work earnestly, and hope that the correction will meet with approval. The manuscript has been overall checked, and the changes marked in red font one by one. We hope that these revisions are sufficient to make our manuscript acceptable for publication in Biosensors. If you have any question about this paper, please do not hesitate to contact me.

Yours sincerely,

Cheng-Te Lin

Ningbo Institute of Material Technology & Engineering, Chinese Academy of Sciences

Round 2

Reviewer 1 Report

Since all the comments were addressed properly. Therefore it can be accepted in the revised form